# *Faecalibacterium duncaniae* Mitigates Intestinal Barrier Damage in Mice Induced by High-Altitude Exposure by Increasing Levels of 2-Ketoglutaric Acid

**DOI:** 10.3390/nu17081380

**Published:** 2025-04-19

**Authors:** Xianduo Sun, Wenjing Li, Guangming Chen, Gaosheng Hu, Jingming Jia

**Affiliations:** School of Traditional Chinese Materia Medica, Shenyang Pharmaceutical University, Shenyang 110016, China; xianduosun2024@163.com (X.S.); wenjingli1886762@163.com (W.L.); 15871473970@163.com (G.C.)

**Keywords:** gut microbiota, 2-ketoglutaric acid, probiotics, gastrointestinal issues, hypoxia exposure

## Abstract

**Background/Objectives:** Exposure to high altitudes often results in gastrointestinal disorders. This study aimed to identify probiotic strains that can alleviate such disorders. **Methods:** We conducted a microbiome analysis to investigate the differences in gut microbiota among volunteers during the acute response and acclimatization phases at high altitudes. Subsequently, we established a mouse model of intestinal barrier damage induced by high-altitude exposure to further investigate the roles of probiotic strains and 2-ketoglutaric acid. Additionally, we performed untargeted metabolomics and transcriptomic analyses to elucidate the underlying mechanisms. **Results:** The microbiome analysis revealed a significant increase in the abundance of Faecalibacterium prausnitzii during the acclimatization phase. *Faecalibacterium duncaniae* (*F. duncaniae*) significantly mitigated damage to the intestinal barrier and the reduction of 2-ketoglutaric acid levels in the cecal contents induced by high-altitude exposure in mice. Immunohistochemistry and TUNEL staining demonstrated that high-altitude exposure significantly decreased the expression of ZO-1 and occludin while increasing apoptosis in ileal tissues. In contrast, treatment with *F. duncaniae* alleviated the loss of ZO-1 and occludin, as well as the apoptosis induced by high-altitude exposure. Furthermore, 2-ketoglutaric acid also mitigated this damage, reducing the loss of occludin and apoptosis in mice. Transcriptomic analysis indicated that high-altitude exposure significantly affects the calcium signaling pathway; conversely, the administration of *F. duncaniae* significantly influenced the PPAR signaling pathway, mineral absorption, and the regulation of lipolysis in adipocytes. Additionally, the expression of the *FBJ osteosarcoma oncogene* (*Fos*) was markedly reduced following the administration of *F. duncaniae*. **Conclusions:** *F. duncaniae* mitigates hypoxia-induced intestinal barrier damage by increasing levels of 2-ketoglutaric acid and shows promise as a probiotic, ultimately aiding travelers in adapting to high-altitude environments.

## 1. Introduction

Many individuals experience gastrointestinal issues when visiting high altitudes [1]. Research indicates that the prevalence of these complaints among travelers to high altitudes via trains is 99%, a rate that significantly exceeds that of other conditions such as hypodynamia (77% of travelers), headaches (74%), and dizziness (71%). Gastrointestinal problems not only occur more frequently but also tend to persist longer and carry greater risks compared with non-gastrointestinal issues [2]. Although the exact causes of gastrointestinal disturbances at high altitudes remain poorly understood, evidence suggests that damage to the intestinal barrier plays a crucial role in their development [1,3]. The intestinal barrier is primarily composed of chemical, biological, and physical barriers [4], all of which can be significantly compromised by high-altitude exposure [5,6,7]. While azithromycin has been reported to effectively reduce high-altitude diarrhea, its efficacy is variable among individuals [8]. Furthermore, several agents, including glutamine [9], vitamin E [10], and butyrate [11], have shown promise in mitigating high-altitude-induced intestinal damage in animal studies. In summary, there are currently very few medications available for alleviating gastrointestinal symptoms associated with high-altitude exposure.

Imbalances in gut microbiota have been linked to the development of gastrointestinal diseases [12]. Research has demonstrated that exposure to high altitudes can alter the composition and structure of gut microbiota. Generally, high-altitude exposure results in an increase in the proportion of anaerobic bacteria and a decrease in facultative anaerobic bacteria [13]. For example, after high-altitude exposure, the abundance of *Odoribacter*, *Prevotella*, and *Akkermansia* increased, while the abundance of *Enterobacter*, *Weissella*, *Escherichia*, and *Klebsiella* decreased [14,15]. Numerous studies have highlighted differences in gut microbiota between low- and high-altitude exposures; however, few studies have specifically examined the differences in gut microbiota during the acute response phase compared with the acclimatization phase at high altitudes. One study noted that travelers transitioning from low to high altitude experienced these two phases, but it did not detail the differences in gut microbiota between them [15]. We believe that investigating these differences could be valuable for identifying microbial factors that contribute to acclimatization at high altitudes. Additionally, the role of gut microbiota altered by high-altitude exposure in gastrointestinal diseases remains a topic of debate. Li et al., 2022 [16] found that high-altitude exposure promoted the growth of Desulfovibrio in the intestines of mice. The phospholipid metabolites produced by Desulfovibrio were then presented by intestinal epithelial Cluster of Differentiation 1D, leading to the proliferation of Interleukin-17A-producing γδ T cells, which further aggravated intestinal injury [16]. In contrast, Su et al., 2024 observed that high-altitude exposure increased the abundance of *Blautia* in the human gut, and this increase was associated with improved gut health and acclimatization at high altitudes [17]. Several factors may contribute to the discrepancies in these findings, including the host species studied, the duration and altitude of exposure, and the specific gut microbes examined in each study. These limited and inconsistent results suggest that much remains to be learned about the role of gut microbiota in intestinal barrier damage caused by high-altitude exposure.

Probiotics, which encompass both bacteria and yeast, are live microorganisms recognized for their beneficial effects on human health. Recently, there has been extensive research focused on probiotics, particularly their potential use as supplemental treatments for various intestinal diseases. Numerous clinical trials have shown that probiotics can positively influence gut microbiota, potentially aiding in the management of various intestinal conditions and promoting overall well-being [18]. Common probiotics include species from the genera *Bacillus*, *Bifidobacterium*, *Clostridium*, *Enterococcus*, *Lactobacillus*, *Saccharomyces*, and *Streptococcus*. Substantial evidence supports the efficacy of these probiotics in treating acute infectious diarrhea, antibiotic-associated diarrhea, *Clostridium difficile*-associated diarrhea, hepatic encephalopathy, ulcerative colitis, irritable bowel syndrome, functional gastrointestinal disorders, and necrotizing enterocolitis [19]. Notably, two prominent probiotics, *Bifidobacterium* and *Lactobacillus*, along with a less common bacterium, *Blautia*, have been shown to alleviate intestinal disorders associated with high altitudes [17,20,21]. Although the range of probiotics studied is limited, their benefits in addressing high-altitude-related intestinal issues warrant further investigation into additional probiotics in this context. This study focuses on probiotics that are adapted to, can survive in, and function effectively within the gut environment created by high-altitude stress, along with their potential mechanisms of action.

In this study, we tested the hypothesis that gut probiotics play a role in mitigating intestinal barrier damage caused by exposure to high altitudes. To investigate this, we conducted microbiome analyses of 49 fecal samples collected from volunteers during both the acute response and acclimatization phases at high altitudes with the aim of identifying potential probiotics. The identified probiotic was subsequently administered via gavage to the intestines of mice to evaluate its protective effects against intestinal barrier damage induced by high-altitude exposure. Additionally, we explored the underlying mechanisms through transcriptome analysis and non-targeted metabolomics. We also examined the beneficial effects of 2-ketoglutaric acid on intestinal barrier damage using mouse models. Our results support the hypothesis that *F. duncaniae* contributes to the reduction in intestinal barrier damage caused by high-altitude exposure by increasing the levels of 2-ketoglutaric acid.

## 2. Materials and Methods

### 2.1. Study Design and Volunteers

This study involved 12 female and 13 male volunteers, all over the age of 20. Participants reported no gastrointestinal abnormalities or diseases, were not taking any medications, and had not undergone a colonoscopy in the past three months. They traveled by train from a low altitude of approximately 100 m to a high altitude of about 3500 m, where they remained for 12 days. On the second day after arriving at the high altitude, the volunteers exhibited one or more symptoms associated with acute mountain sickness (AMS). On this day, a total of 25 fresh stool samples were collected and categorized as the unacclimatized group (acute response phase). After 12 days at high altitude, all AMS-related symptoms had nearly resolved, and an additional 24 fresh stool samples were collected on this day and classified as the acclimatized group (acclimatization phase). All fecal samples were stored at −80 °C for 16S rRNA sequencing.

### 2.2. Microbiome Analysis

Fecal samples collected from volunteers were immediately stored at −80 °C until microbiome analysis was conducted at Shanghai Majorbio Technology Co., Ltd. (Shanghai, China). The microbiome analysis was performed as previously described [22], with some modifications; specifically, we calculated rarefaction curves and alpha diversity indices, including Chao1 richness and the Shannon index, using Mothur v1.30.2 [23] based on the Operational Taxonomic Units (OTUs). To assess the similarities between microbial communities across different samples, we employed Partial Least Squares Discriminant Analysis (PLS-DA) using R (version 3.3.1). Additionally, the microbial dysbiosis index was utilized to evaluate the state of microbial dysbiosis.

### 2.3. Optimal Prebiotic Analysis of F. duncaniae

*F. duncaniae* was obtained from Ningbo Testobio Biotechnology Co., Ltd. (Ningbo, China) To cultivate *F. duncaniae* anaerobically, a modified reinforced clostridial broth was used, containing the following components: tryptose (10 g/L), beef extract (10 g/L), yeast extract (3 g/L), dextrose (5 g/L), NaCl (5 g/L), soluble starch (1 g/L), L-cysteine hydrochloride (0.5 g/L), and sodium acetate (3 g/L). The pH was maintained at 6.8 ± 0.2. Cultivation was conducted at 37 °C. Prior to use, all broths were treated to remove oxygen, ensuring an anaerobic environment for the culture. *F. duncaniae* strains in the logarithmic growth phase were diluted to an OD600 of 0.4. Subsequently, 10% of the diluted broth was added to two types of modified reinforced clostridial broth: one without dextrose and the other supplemented with one of the following carbon sources: 5 g/L of inulin (Yuanye Biotechnology, Shanghai, China), soybean oligosaccharide (Yuanye Biotechnology, Shanghai), fructooligosaccharide (Yuanye Biotechnology, Shanghai), galactooligosaccharide (Yuanye Biotechnology, Shanghai), konjac glucomannan (Yuanye Biotechnology, Shanghai), isomaltooligosaccharide (Yuanye Biotechnology, Shanghai), or dextrose (Yuanye Biotechnology, Shanghai). The broth was incubated at 37 °C for 24 h, with three replicates for each group. At the end of the incubation period, absorbance values were measured. Differences in absorbance values between the groups with various carbon sources and the group without dextrose were calculated to investigate the effects of different prebiotics on the growth of *F. duncaniae*.

### 2.4. Preparation of Probiotics for Gavage

*F. duncaniae* was cultured anaerobically in modified reinforced clostridial broth for 24 h at 37 °C. Following the cultivation period, the bacterial precipitate was collected via centrifugation and washed twice with sterile anaerobic PBS. The concentration of the strains was adjusted using sterile anaerobic PBS containing 20 mg/mL of inulin, resulting in a synbiotic with approximately 5 × 10^9^ CFU/mL of *F. duncaniae* [24].

### 2.5. Animal Experiment

Healthy male C57BL/6 mice aged six weeks were obtained from Liaoning Changsheng Biotech Co., Ltd. (Shenyang, China). They were housed in a controlled environment with a temperature of 20 ± 5 °C and a relative humidity of 50 ± 5% at a low altitude (below 500 m). The mice were maintained on a 12-h light/dark cycle and had unlimited access to food and water. Prior to the commencement of the experiment, all mice underwent a seven-day acclimation period.

To determine the optimal duration for modeling, the mice were randomly divided into three groups: a normoxia control (NC) group, a hypoxia exposure group for 5 days (H5D), and a hypoxia exposure group for 10 days (H10D). Each group consisted of five mice. The NC group received sterile PBS administered orally once daily and remained in a normoxic environment for the entire 10 days. The H5D group was kept in a normoxic environment for the first 5 days, followed by 5 days in a hypobaric chamber (Yuyan Instruments, Shanghai, China) at a simulated altitude of 6500 m, while also receiving sterile PBS administered orally once daily. The H10D group received sterile PBS administered orally once daily and remained in the hypobaric chamber at a simulated altitude of 6500 m for 10 days.

To investigate the effects of *F. duncaniae* on intestinal barrier damage caused by high-altitude exposure, the mice were randomly divided into three groups: the normoxia control group (NC), the hypoxia group (H), and the hypoxia + *F. duncaniae* and inulin group (HF). Each group comprised eight mice. The mice in the NC group received sterile anaerobic PBS administered orally once daily and remained in a normoxic environment for 17 days. The mice in the H group received sterile anaerobic PBS administered orally once daily and were housed in a normoxic environment for the first seven days, followed by 10 days in a hypobaric chamber at a simulated altitude of 6500 m, continuing to receive sterile anaerobic PBS administered orally once daily. The mice in the HF group were treated with *F. duncaniae* and inulin once daily, remaining in a normoxic environment for seven days, followed by 10 days in the hypobaric chamber at a simulated altitude of 6500 m, during which they continued to receive *F. duncaniae* and inulin administered orally once daily.

In the experiment involving treatment with 2-ketoglutaric acid, mice were randomly divided into three groups after a period of acclimatization: the normoxia control (NC) group, the hypoxia (H) group, and the hypoxia + 2-ketoglutaric acid (H2K) group. Each group comprised eight mice. In the NC group, the mice were administered sterile PBS orally once daily and kept in a normoxic environment for 10 days. The mice in the H group also received sterile PBS administered orally once daily but were housed in a hypobaric chamber simulating an altitude of 6500 m for the same duration. The mice in the H2K group were treated with 10 mg/kg of 2-ketoglutaric acid [25] via gavage once daily, received 0.05% 2-ketoglutaric acid in their drinking water, and remained in the hypobaric chamber at a simulated altitude of 6500 m for 10 days. The gavage volume for all the mouse experiments was set at 10 mL/kg. After the treatment period, all mice were euthanized under isoflurane anesthesia, and relevant samples were collected for subsequent analyses.

### 2.6. Real-Time PCR Assay

Cecal contents were collected and stored at −80 °C for the quantitative analysis of *F. duncaniae* abundance. DNA was extracted from the cecal contents using the FastPure Stool DNA Isolation Kit (MJYH, Shenzhen, China) according to the manufacturer’s instructions. The qPCR primer sets used are listed in Appendix A. Real-time PCR (RT-PCR) assays were conducted in 96-well optical plates on an ABI 7300 fluorescent quantitative PCR instrument (Applied Biosystems, Waltham, MA, USA). The reaction mixture consisted of 5 μL of 2× ChamQ SYBR Color qPCR Master Mix (Nuoweizan, Nanjing, China), 1 μL of template DNA, 0.4 μL of 5 μM forward primer, 0.4 μL of 5 μM reverse primer, 0.2 μL of 50× ROX Reference Dye 1, and 3 μL of double-distilled H_2_O. The PCR cycling conditions included an initial denaturation at 95 °C for 3 min, followed by 40 cycles of melting at 95 °C for 5 s, annealing at 58 °C for 30 s, and a final extension at 72 °C for 1 min. The quantity of target DNA was determined by comparing it with serially diluted standards tested on the same plate. Bacterial abundance was expressed as log_10_ copies per gram of cecal contents.

Total RNA was isolated from ileum tissues using the MJZol total RNA extraction kit (Majorbio, Shanghai, China) to verify the expression of the *Nfkbia* and *Fos* genes. cDNA was reverse transcribed using the HiScript Q RT SuperMix for qPCR (+gDNA wiper) Synthesis Kit (Vazyme Biotech Co., Ltd., Nanjing, China) following the manufacturer’s instructions. RT-PCR was performed in triplicate for each gene, and the GAPDH gene was used as an internal standard. All paired primers used for RT-PCR are listed in Appendix A. RT-PCR assays were conducted on a QuantStudio^TM^ 5384-well real-time PCR system (Applied Biosystems, Waltham, MA, USA). The reaction mixture comprised 5 μL of 2× ChamQ SYBR Color qPCR Master Mix (Nuoweizan, Nanjing, China), 1 μL of template cDNA, 0.4 μL of 5 μM forward primer, 0.4 μL of 5 μM reverse primer, 0.2 μL of 50× ROX Reference Dye 2, and 3 μL of double-distilled H_2_O. The PCR cycling conditions included an initial denaturation at 95 °C for 5 min, followed by 35 cycles of melting at 95 °C for 30 s, annealing at 55 °C for 30 s, and a final extension at 72 °C for 1 min. Relative transcription levels were quantified using the 2^−ΔΔCT^ method.

### 2.7. Intestinal Permeability Analysis

Intestinal permeability was assessed using the 3–5 kDa fluorescein isothiocyanate (FITC)-dextran test (BioDuly, Nanjing, China), following previously established methods [26]. In summary, FITC-dextran (600 mg/kg body weight) was administered to mice via oral gavage four hours prior to sacrifice. Subsequently, sera were obtained by centrifuging the samples at 2500 rpm for 20 min at 4 °C. A volume of 100 μL of each serum sample was transferred to a 96-well microplate. If the serum collected from an individual mouse was less than 100 μL, sera from two mice in the same group were combined to form a single sample. Fluorescence was measured at an excitation wavelength of 485 nm and an emission wavelength of 528 nm, and the concentration of FITC-dextran was calculated using a standard curve (y = 2.1085x − 0.4255, R^2^ = 0.9997).

### 2.8. H&E and PAS Staining

A portion of the ileum was collected and preserved in 4% paraformaldehyde (Solarbio, Beijing, China). The sample was embedded in paraffin and sectioned into 3–4 μm slices. These slices were stained using the H&E Staining Kit (Servicebio, Wuhan, China) and the PAS dye solution set (Servicebio, Wuhan, China), following the manufacturer’s specifications. The stained sections were examined under a light microscope. Pathological changes were assessed by measuring the height of the intestinal villi and the thickness of the mucosa. Villi height was measured from the neck of the crypt to the tip of the villus using IMAGEJ, version 1.80, with a minimum of 15 villi measured for each sample. The number of goblet cells was evaluated in non-consecutive, randomly chosen histological fields (5 per slide). Two experienced pathologists, blinded to group identities, conducted pathological examinations of the specimens.

### 2.9. Immunohistochemistry Staining

Paraffin sections were deparaffinized in water, and antigen retrieval was performed using a Tris-EDTA buffer in a microwave oven. After cooling, the sections were washed three times with PBS (pH 7.4), with each wash lasting 5 min. They were then incubated in 3% hydrogen peroxide for 25 min to quench endogenous peroxidase activity, followed by three additional washes with PBS. Next, the sections were blocked with 3% BSA for 30 min. The sections were incubated overnight at 4 °C with primary antibodies against ZO-1 (GB115686, 1:500, Servicebio) or occludin (GB111401, 1:500, Servicebio). After washing with PBS three times, the sections were incubated for 50 min with an HRP-labeled goat anti-rabbit IgG secondary antibody (GB23303, 1:200, Servicebio). Following this, DAB stain (Servicebio, Wuhan, China) was added for color development, and the nuclei were re-stained with hematoxylin (Servicebio, Wuhan, China). The sections were then sealed with Rhamsan gum (Servicebio, Wuhan, China). Finally, images were captured and recorded using a Nikon Eclipse E100 microscope (Nikon, Tokyo, Japan), and quantitative analysis of the images was performed using the Aipathwell v2 software (ServiceBio, Wuhan, China).

### 2.10. TUNEL Staining

Apoptotic cells in the ileum tissues were detected using a TUNEL assay kit (Servicebio, G1501, Wuhan, China) according to the manufacturer’s instructions. Fluorescence signals were observed and recorded with a NIKON ECLIPSE C1 Fluorescence Microscope. Quantitative analysis was performed using the Aipathwell software (ServiceBio, Wuhan, China).

### 2.11. Untargeted Metabolomics Analysis

Untargeted metabolomics analysis was conducted at Shanghai Majorbio Technology Co., Ltd. Metabolite extraction from cecum contents was performed as previously described [27]. LC-MS/MS analysis utilized a SCIEX UHPLC-Triple TOF 6600 system (SCIEX, Framingham, MA, USA), employing the same column and mobile phases as outlined earlier [27]. However, in this study, the column temperature was set to 45 °C. The UPLC system was coupled to a quadrupole-time-of-flight mass spectrometer equipped with an electrospray ionization (ESI) source, operating in both positive and negative modes. The optimal mass spectrometry conditions for this study were as follows: source temperature at 500 °C; curtain gas at 35 psi; ion source gas 1 at 50 psi and gas 2 at 50 psi; ion-spray voltage floating at −4500 V in negative mode and 5500 V in positive mode; declustering potential at 80 V; and collision energy set to 40 ± 20 eV for MS/MS. Data acquisition was performed using the Information Dependent Acquisition mode, and detection occurred over a mass range of 50–1200 *m*/*z*. The processing of raw LC/MS data and identification of metabolites were completed as previously described [27]. The data matrix obtained from the database search was uploaded to the Majorbio cloud platform for data analysis [27]. The R package “ropls” (Version 1.6.2) was used to conduct Partial Least Squares Discriminant Analysis (PLS-DA) and Orthogonal Partial Least Squares Discriminant Analysis (OPLS-DA). This analysis included a 7-cycle interactive validation to assess model stability. Metabolites with a Variable Importance in Projection (VIP) greater than 1 and a *p*-value less than 0.05 were identified as significantly different based on the results from the OPLS-DA model and Student’s *t*-test. Additionally, pathway analysis using the KEGG database was performed to map these differential metabolites between the two groups to their relevant biochemical pathways.

### 2.12. Measurement of AST and IDH Activities, as Well as the Levels of 2-Ketoglutaric Acid and Total Protein

The enzymatic activities of aspartate aminotransferase (AST) and isocitrate dehydrogenase (IDH) in the cecal contents were measured using an Aspartate Aminotransferase Assay Kit (Nanjing Jiancheng Bioengineering Institute, Nanjing, China) and an Isocitrate Dehydrogenase Activity Assay Kit with WST-8 (Beyotime, Haimen, China), respectively. The total protein concentration in the cecal contents was determined using a BCA Protein Assay Kit (BCM Biotech, Shanghai, China). Additionally, the levels of 2-ketoglutaric acid in the cecal contents were assessed with an Amplex Red α-Ketoglutarate Assay Kit (Beyotime, Haimen, China). All the procedures were performed following the instructions provided in the reagent kits, and the results were read using an ELX800 universal microplate reader (Bio-Tek Instruments Inc., Winooski, VT, USA).

### 2.13. Transcriptome Analysis

Total RNA extraction from the ileum, library preparation, sequencing, quality control, and read mapping were conducted at Shanghai Majorbio Bio-pharm Biotechnology Co., Ltd., as previously described [28]. Transcript expression levels were calculated using the transcripts per million reads (TPM) method. Gene abundance was quantified with RSEM, and differential expression analysis was performed using DESeq2. Genes with a log_2_ fold change (|log_2_FC|) ≥ 2 and a false discovery rate (FDR) ≤ 0.05 were considered significantly differentially expressed genes (DEGs). Additionally, functional enrichment analysis, including Gene Ontology (GO) and Kyoto Encyclopedia of Genes and Genomes (KEGG), was performed to identify DEGs that were significantly enriched in specific GO terms and metabolic pathways. GO functional enrichment and KEGG pathway analyses were carried out using Goatools (Version 1.4.4) and KOBAS (Version 2.1.1), respectively.

### 2.14. Statistical Analysis

Statistical analysis was conducted using SPSS version 19.0 software, with data presented as mean ± standard deviation. For data following a normal distribution, comparisons were made using a two-tailed Student’s *t*-test or a one-way ANOVA. In cases of non-parametric distributions, the Wilcoxon rank-sum test was employed for group comparisons.

## 3. Results

### 3.1. Differences in Gut Microbiota Between Volunteers During Acute Response and Acclimatization Phases at High Altitude

To identify gut microbiota potentially involved in human acclimatization to high altitudes, we categorized fecal samples collected from volunteers transitioning from low to high altitude into two groups: unacclimatized (Una) and acclimatized (A). First, we assessed the adequacy of the sample size. As illustrated in Figure 1A,B, the Pan/Core species curve began to plateau as the number of samples increased, indicating that the sample size was sufficient for sequencing. Additionally, the rarefaction curves based on the Sobs and Shannon indices (Figure 1C,D) demonstrated that as the number of sequences increased, the curves gradually leveled off, indicating saturated coverage and sufficient sequencing depth for the experiment. We also analyzed alpha diversity and found no significant differences between the Una and A groups based on the CHAO (Figure 1E) and SHANNON (Figure 1F) indices. However, the microbial dysbiosis index (MDI) was significantly higher in the Una group than in the A group (Figure 1G). Furthermore, beta diversity analysis revealed a distinct separation between the samples in the Una and A groups (Figure 1H). These results indicate significant differences in gut microbiota between the Una and A groups. Analysis of species composition revealed that the most abundant genera in both groups were primarily *Prevotella*, *Faecalibacterium*, *Bacteroides*, *Bifidobacterium*, and *Blautia* (Figure 1I). Generally, a higher abundance of these genera suggests a greater likelihood of influencing host health or disease.

To identify the significantly different microbial species between the Una and A groups, we conducted a linear discriminant analysis effect size (LEfSe) analysis. At the phylum and family levels, we found that *Proteobacteria*, *Enterobacteriaceae*, *Morganellaceae*, *Actinomycetaceae*, *Leuconostocaceae*, *Vagococcaceae*, *Barnesiellaceae*, and *Enterococcaceae* were significantly more abundant in the Una group (Figure 1J). At the genus and species levels, the following were significantly enriched in the Una group: *Escherichia-Shigella*, *Bacteroides plebeius*, *Proteus*, *Solobacterium moorei*, *Weissella*, *Actinomyces*, *Bacteroides proteus*, *Schaalia odontolytica*, *Vagococcus*, *Barnesiella*, and *Lactococcus*. In contrast, *Faecalibacterium prausnitzii* (*F. prausnitzii*), *Lachnospira*, and Subdoligranulum were significantly more abundant in the A group (Figure 1K). Most of the bacteria enriched in the Una group were facultative anaerobes and opportunistic pathogens, while those in the A group consisted of obligate anaerobes and potentially beneficial bacteria.

### 3.2. Evaluation of the Most Appropriate Prebiotics for F. duncaniae

Prebiotics are known to support the growth of probiotics in the host gut. To enhance the survival and beneficial role of *F. duncaniae* in the murine gut, we investigated the effects of six common prebiotics on the growth of *F. duncaniae*. Our results demonstrated that inulin exhibited the most significant proliferative effect on *F. duncaniae*, surpassing glucose and the other prebiotics. Following inulin, the effectiveness of the remaining prebiotics was ranked as follows: soybean oligosaccharide, galactooligosaccharide, konjac glucomannan, fructooligosaccharide, and isomaltooligosaccharide (Appendix A). Therefore, we selected inulin as the most effective prebiotic for *F. duncaniae*.

### 3.3. F. duncaniae Mitigates Intestinal Barrier Damage Induced by High-Altitude Exposure

To determine the optimal duration for modeling, we divided mice into three groups: a normoxia control group (NC), a 5-day hypoxia exposure group (H5D), and a 10-day hypoxia exposure group (H10D). We assessed intestinal permeability after the treatment period. The results indicated that intestinal permeability in the H10D group was significantly higher than in both the NC and H5D groups. Thus, we concluded that a 10-day hypoxia exposure is optimal for inducing intestinal barrier injury in mice (Figure 2A).

To evaluate whether *F. duncaniae* possesses a protective effect against hypoxia-induced damage to the intestinal barrier, we divided the mice into three groups: a normoxia control group (NC), a hypoxia group (H), and a hypoxia + *F. duncaniae* and inulin group (HF) (Figure 2B). We initially measured the abundance of *F. duncaniae* in the cecal contents using a real-time PCR assay. The results demonstrated that the abundance of *F. duncaniae* in the HF group was significantly higher than that in the other groups (Figure 2C). Subsequently, we recorded the weights of the mice every two days. Following hypoxia exposure, the weights of the mice in both the H and HF groups decreased significantly. Although the HF group exhibited a higher body weight than the H group, the difference was not statistically significant (Figure 2D).

Regarding H&E staining, we made the following observations: In the NC group, the structure of all layers of the ileum remained intact, displaying a normal shape with no pathological changes. In the H group, the lamina propria of the ileum exhibited areas of mild edema, a loose arrangement of connective tissue, and widening glandular septa, accompanied by a small amount of lymphocyte infiltration (blue arrow). Additionally, a considerable portion of the intestinal gland structure in the mucosal layer was absent, with a significant presence of lymphoid tissue (red arrow) and an increased number of lymphocytes. The surrounding intestinal glands exhibited an irregular arrangement (green arrow). In the HF group, the mucosal structure was intact, with abundant and closely arranged lamina propria intestinal glands. The myocytes of the myometrium were regularly arranged and displayed normal morphology (Figure 2E). Furthermore, compared with the NC group, the H group showed significant reductions in villus height, crypt depth, and the number of goblet cells per villus. After treatment with *F. duncaniae*, all three parameters significantly increased (Figure 2F,J–L).

Immunohistochemical analysis was conducted to assess the levels of ZO-1 and occludin in ileal tissues. The results indicated a significant decrease in the expression of ZO-1 and occludin in the H group, while there was a notable increase in the HF group, bringing their levels close to those observed in the NC group (Figure 2G,H,M,N). These findings suggest that *F. duncaniae* effectively mitigates the loss of ZO-1 and occludin induced by high-altitude exposure, potentially restoring intestinal barrier function. We also evaluated apoptosis in the ileal tissues. The TUNEL assay showed minimal green fluorescence in both the NC and HF groups, while a substantial increase in apoptosis was detected in the H group (Figure 2I,O). Additionally, we measured intestinal permeability by analyzing serum FD-4kDa levels. The results revealed that serum FD-4kDa levels in the H group were significantly higher than those in the NC group. However, the serum FD-4kDa levels in the HF group were significantly lower than those in the H group (Figure 2P), indicating that high-altitude exposure increases intestinal permeability, while *F. duncaniae* supplementation may reverse this increase.

### 3.4. Influence of F. duncaniae on the Metabolome of Cecal Contents

To clarify the potential role of intestinal metabolites in the development and recovery of intestinal barrier damage caused by high altitude, we conducted untargeted metabolomics analyses on cecal contents from mice. We employed PLS-DA, a supervised algorithm that integrates feature extraction with discriminant analysis. The PLS-DA results demonstrated that samples from the NC and H groups, as well as samples from the H and HF groups, were distributed in distinct areas and could be completely separated. In contrast, samples within the same groups clustered together (Figure 3A,B). This indicates that the measured metabolic data exhibited good repeatability, with clear differences among the various groups. An enhanced volcano plot revealed a total of 212 significantly different metabolites between the H and NC groups. Of these, 101 metabolites were significantly elevated, while 111 were significantly decreased in the H group (Figure 3C). In comparing the HF and H groups, we identified 207 significantly different metabolites, with 131 significantly increased and 76 significantly decreased in the HF group (Figure 3D). Utilizing the weighting coefficient from the OPLS-DA model, we ranked the contributions of differential metabolites to the discrimination of the various groups based on their VIP scores. The top 50 contributing metabolites between the NC and H groups are displayed in Figure 3E, while the top 20 contributing metabolites between the H and HF groups are presented in Figure 3F.

Additionally, we performed KEGG pathway enrichment analysis, which revealed that the differential metabolites between the NC and H groups were significantly enriched in several pathways, including the regulation of lipolysis in adipocytes, thermogenesis, biosynthesis of unsaturated fatty acids, linoleic acid metabolism, nucleotide metabolism, amoebiasis, and the HIF-1 signaling pathway (Figure 3G). Further analysis identified four differential metabolites closely associated with the HIF-1 signaling pathway: Dg(24:1(15Z)/15:0/0:0), ascorbic acid, 2-ketoglutaric acid, and Dg(22:4(7Z,10Z,13Z,16Z)/20:1(11Z)/0:0) (Figure 3H–K). Notably, the abundance of 2-ketoglutarate was significantly lower in the H group compared with the NC group, while the abundance of 2-ketoglutaric acid in the HF group was significantly higher than that in the H group (Figure 3J). We quantified the level of 2-ketoglutaric acid in cecal contents using an assay kit, and the results indicated that the level in the H group was significantly lower than in the NC group. Moreover, the level of 2-ketoglutaric acid in the HF group was significantly higher than in the H group (Figure 3L). This finding is consistent with the results of the untargeted metabolomics analyses. Furthermore, we measured the activity levels of AST and IDH in the cecal contents. The results indicated that AST activity in the HF group was significantly higher than that in the H group, and IDH activity was significantly elevated in both the HF and NC groups compared with the H group (Figure 3M,N).

### 3.5. Influence of F. duncaniae on the Transcriptome of Ileal Tissues

To investigate the transcriptional regulatory mechanisms through which high-altitude exposure induces intestinal barrier damage and how *F. duncaniae* alleviates this damage at the molecular level in mice, we sequenced ileal tissues from mice to obtain transcriptome profiles. We identified differentially expressed genes (DEGs) from the transcriptome data. The volcanic plot indicated that, compared with the NC group, a total of 301 DEGs were identified in the H group, with 187 upregulated and 114 downregulated (Figure 4A). Additionally, when comparing the HF group to the H group, we found a total of 267 DEGs, including 125 that were upregulated and 141 that were downregulated (Figure 4B). We further conducted GO and KEGG enrichment analyses using the DEGs. The most significantly enriched GO pathways when comparing the NC and H groups included Extracellular Space, Immunoglobulin Complex, and Immune System Process (Figure 4C). The most significantly enriched KEGG pathways when comparing the NC and H groups were Malaria, African Trypanosomiasis, and Neuroactive Ligand–Receptor Interaction (Figure 4E). Additionally, eight DEGs were significantly enriched in the Calcium Signaling Pathway. In the comparison between the HF and H groups, the most significantly enriched GO pathways included the Immunoglobulin Complex, Circulating Immunoglobulin Receptor Binding, Phagocytosis, Recognition, Complement Activation, and Classical Pathway (Figure 4D). The most significantly enriched KEGG pathways identified were the PPAR Signaling Pathway, Mineral Absorption, and Regulation of Lipolysis in Adipocytes (Figure 4F). Furthermore, two DEGs (*Nfkbia* and *Fos*) showed enrichment in the Apoptosis Pathway, although this was not statistically significant. The DEGs associated with the Calcium Signaling Pathway and Apoptosis Pathway are displayed in heat maps (Figure 4G). We validated the expression of *Nfkbia* and *Fos* using RT-qPCR. The results indicated that the expression of *Nfkbia* was higher in the H group compared with both the NC and HF groups, although this difference was not statistically significant (Figure 4H). Similarly, the expression of *Fos* was greater in the H group than in the NC group, but again, the difference was not significant. In contrast, when comparing the H group to the HF group, the expression of *Fos* was significantly decreased in the HF group (Figure 4I).

### 3.6. 2-Ketoglutaric Acid Aids in the Repair of Intestinal Barrier Damage Induced by High-Altitude Exposure in Mice

To investigate the protective effect of 2-ketoglutaric acid on intestinal barrier damage caused by high-altitude exposure in mice, we divided the mice into three groups: a normoxia control (NC) group, a hypoxia (H) group, and a hypoxia + 2-ketoglutaric acid (H2K) group (Figure 5A). The experimental results indicated that after high-altitude exposure, the body weights of mice in both the H and H2K groups significantly decreased compared with the body weights of mice in the NC group. However, there was no significant difference in body weight between the H and H2K groups (Figure 5B).

Histological analysis using H&E staining revealed the following observations. In the NC group, the structure of the ileum remained intact, exhibiting a normal shape with no pathological changes observed. In the H group, the lamina propria of the ileum displayed a substantial area of mildly loose connective tissue, accompanied by scattered lymphocytes (yellow arrow), visible lymph nodes (black arrow), and rare instances of extravasated blood (red arrow). Additionally, occasional absence of mucosal epithelial cells was noted (blue arrow). In the H2K group, the lamina propria of the ileum also exhibited mildly loose connective tissue (green arrow) and rare occurrences of extravasated blood (red arrow). Instances of mild edema in the mucosal epithelial cells were observed (white arrow), but no other significant abnormalities were detected (Figure 5D). Furthermore, compared with the NC group, the H group showed significant reductions in villus height, crypt depth, and the number of goblet cells per villus. Following treatment with 2-ketoglutaric acid, both villus height and crypt depth were significantly increased (Figure 5C,H,I).

Immunohistochemical analysis indicated a significant decrease in the expression of occludin in the H group compared with the NC group. However, treatment with 2-ketoglutaric acid resulted in a notable increase in the expression of this protein (Figure 5F,J). These findings suggest that 2-ketoglutaric acid effectively mitigates the loss of occludin induced by high-altitude exposure, potentially restoring intestinal barrier function. Additionally, the TUNEL assay revealed that high-altitude exposure resulted in significant apoptosis in the ilea of mice in the H group. In contrast, treatment with 2-ketoglutaric acid significantly reduced this apoptosis (Figure 5G,K). We also assessed intestinal permeability. The results showed that serum FD-4kDa levels in the H group were significantly higher than those in the NC group. However, serum FD-4kDa levels in the H2K group were significantly lower than those in the H group (Figure 5L). These results suggest that 2-ketoglutaric acid alleviates intestinal barrier damage resulting from high-altitude exposure in mice.

## 4. Discussion

Common probiotics are widely recognized for their role in supporting gut health. However, exposure to high altitudes significantly alters the intestinal microenvironment, which may hinder the survival of certain common probiotics [29]. Therefore, it is crucial to identify specific gut microbiota that can adapt to these changes and play a probiotic role, particularly in preventing and treating intestinal barrier damage at high altitudes. Investigating the differences in gut microbiota between acculturated and unacculturated individuals living at high altitudes may help uncover these specific microorganisms. In our study, we examined gut microbiota during the acute response and acclimatization phases in volunteers who traveled from low to high altitude. We found that many opportunistic pathogens significantly increased during the acute response phase, leading us to speculate that these pathogens may be associated with human inadaptation to high-altitude conditions. Notably, we observed a significant increase in the abundance of *F. prausnitzii* during the acclimatization phase. Research indicates that *F. prausnitzii* is the most prevalent bacterium in the intestines of healthy adults, comprising over 5% of the total bacterial population. Administration of the *F. prausnitzii* strain A2-165, along with its culture supernatant, has been shown to protect against colitis induced by 2,4,6-trinitrobenzenesulfonic acid in mice [30]. This protective effect is primarily attributed to its potent anti-inflammatory properties [30]. Additionally, supplementation with *F. prausnitzii* can prevent intestinal barrier damage associated with sleep deprivation or acute myeloid leukemia, a benefit linked to the bacterium’s butyrate production [31,32]. Recently, new species of *Faecalibacterium* have been identified and some strains of *F. prausnitzii* isolated from human feces, including the reference strain A2-165, have been reclassified as *F. duncaniae* [33,34]. However, the role and mechanisms of action of *Faecalibacterium* in intestinal damage caused by high altitudes remain unclear.

Exposure to high altitudes often results in gastrointestinal issues, although the exact causes are not fully understood. One possible explanation is that hypoxia can damage the intestinal barrier [1]. Research has shown that rodents exposed to hypoxia experience significant injury to epithelial cells and a compromised intestinal barrier, resulting in increased permeability and bacterial translocation [1]. In a study by Karl et al., 2018, increased intestinal permeability was observed in a group of 17 healthy, physically active, but unacclimatized men after a rapid 22-day exposure to an altitude of 4300 m [35]. Our findings also demonstrated that high-altitude exposure induced intestinal barrier damage in mice, consistent with previous research. Interestingly, we discovered that supplementation with *F. duncaniae* in mice reduced gut barrier damage and decreased permeability. These results suggest that *F. duncaniae* may function as a probiotic bacterium that helps mitigate intestinal barrier damage caused by high altitude and could facilitate human adaptation to high-altitude environments. Additionally, we observed a significant increase in the relative abundance of several other bacterial species, such as *Lachnospira*, *Subdoligranulum*, and *Anaerostipes caccae* DSM 14662, during the acclimatization phase compared with the acute response phase. Previous studies have indicated that *Subdoligranulum* is beneficial for alleviating necrotizing enterocolitis [36], while *Anaerostipes caccae* is recognized as an important butyrate-producing bacterium [37]. This information suggests that studying the synergistic effects of these bacteria and *F. duncaniae* in alleviating intestinal diseases induced by high-altitude exposure holds great potential for future applications.

Previous studies have often linked the probiotic effects of *Faecalibacterium* to its ability to produce butyric acid [31,32], while other active components have frequently been overlooked. In this study, we observed that exposure to high altitude significantly reduced the levels of 2-ketoglutaric acid in cecal contents. However, supplementation with *F. duncaniae* resulted in a notable recovery of 2-ketoglutaric acid levels. These findings suggest that restoring 2-ketoglutaric acid levels may be beneficial. As an intermediate in the tricarboxylic acid (TCA) cycle, 2-ketoglutaric acid plays a crucial role in energy metabolism as well as the metabolism of carbon and nitrogen, underscoring its multiple important functions. Within the digestive system, 2-ketoglutaric acid contributes to the TCA cycle, providing energy for intestinal epithelial cells and helping to maintain intestinal mucosal integrity [38]. Previous research has demonstrated that 2-ketoglutaric acid reduces the production of reactive oxygen species, inflammation, and apoptosis [38,39]. Notably, exposure to high altitude is associated with limited energy metabolism, increased oxidative stress, elevated inflammation, and heightened apoptosis, all strongly linked to altitude-induced diseases [40]. In our study, we found that additional supplementation of 2-ketoglutaric acid alleviated hypoxia-induced intestinal barrier damage and reduced intestinal cell apoptosis. Due to its multiple functions, 2-ketoglutaric acid is recognized as a safe nutritional supplement in clinical applications, exhibiting protective effects against various diseases, including aging, muscle mass loss, osteoporosis, neurodegenerative diseases, and cardiovascular diseases [41]. However, the role of 2-ketoglutaric acid in treating gastrointestinal disorders induced by high-altitude exposure has not been previously established. The results of this study suggest that 2-ketoglutaric acid holds significant potential as a nutritional supplement for the treatment or prevention of gastrointestinal disorders associated with high-altitude exposure. Additionally, to explore the mechanism through which *F. duncaniae* restores 2-ketoglutaric acid levels, we examined KEGG databases and found that it can express AST and IDH enzymes. Both of these enzymes catalyze specific substrates to produce 2-ketoglutaric acid. Our analysis demonstrated that supplementation with *F. duncaniae* significantly enhanced the activity of these two enzymes in cecal contents.

We analyzed transcriptome data to investigate the mechanisms underlying intestinal barrier damage caused by high-altitude exposure and how *F. duncaniae* may mitigate this damage at the molecular level. GO enrichment analysis revealed that many DEGs were significantly enriched in various immune-related pathways, suggesting a link between the immune system and intestinal barrier damage resulting from high-altitude exposure. Interestingly, 2-ketoglutaric acid, known as an immune nutrient factor, plays a crucial role in general immune metabolism. As a glutamine homolog, 2-ketoglutaric acid possesses immuno-enhancing properties, maintaining the gut barrier, increasing the activity of immune cells and neutrophils, promoting phagocytosis, and reducing bacterial translocation in vivo [42]. We hypothesize that *F. duncaniae* may modulate mouse immunity by restoring ketoglutaric acid levels, which could be beneficial in protecting the intestinal barrier from damage caused by high-altitude exposure. Furthermore, KEGG enrichment analysis indicated that eight DEGs between the NC and H groups were significantly enriched in the Calcium Signaling Pathway. Previous studies have shown that hypoxia can disrupt intracellular calcium levels [43], potentially leading to calcium overload, mitochondrial uncoupling, decreased ATP synthesis, and, ultimately, cell death [44]. Additionally, the calcium signaling pathway plays a critical role in regulating cell apoptosis [45,46]. Therefore, we speculate that hypoxia induces intestinal cell apoptosis through the calcium signaling pathway, exacerbating intestinal barrier damage. In contrast, DEGs between the HF and H groups exhibited significant enrichment in several pathways, including the PPAR Signaling Pathway, Mineral Absorption, Regulation of Lipolysis in Adipocytes, and Arachidonic Acid Metabolism. Notably, the PPAR Signaling Pathway, Regulation of Lipolysis in Adipocytes, and Arachidonic Acid Metabolism are all associated with fat metabolism, suggesting that fat metabolism may play a crucial role in preventing intestinal barrier damage, highlighting a valuable area for future research. The Mineral Absorption pathway is associated with calcium absorption and may contribute to regulating calcium homeostasis within cells. Interestingly, we identified two DEGs, *Nfkbia* and *Fos*, that were enriched in the apoptosis pathway, with their abundance significantly decreasing after treatment with *F. duncaniae*. Inhibition of *Nfkbia* and *Fos* expression has been shown to promote cell survival [47,48,49]. Thus, we speculate that *F. duncaniae* may mitigate intestinal barrier damage caused by high-altitude exposure by regulating both calcium homeostasis and apoptosis.

## 5. Conclusions

In conclusion, our results indicate that exposure to high altitudes compromises the intestinal barrier in mice. However, supplementation with *F. duncaniae* effectively mitigates this damage. Additionally, we observed a significant decrease in the levels of 2-ketoglutaric acid in the cecal contents following exposure to high altitude; nevertheless, *F. duncaniae* supplementation reversed this decrease. Notably, supplementation with 2-ketoglutaric acid also alleviated the damage in the mice. Regarding the underlying mechanisms, we speculate that *F. duncaniae* may help prevent the decline in 2-ketoglutaric acid levels by enhancing the activities of AST and IDH in the intestines. The recovery of 2-ketoglutaric acid levels promotes energy metabolism in intestinal cells and reduces apoptosis by modulating calcium signaling and apoptosis pathways, thereby helping to mitigate intestinal barrier damage caused by high altitudes. This study suggests that *F. duncaniae* has potential as a probiotic for alleviating gastrointestinal discomfort at high altitudes and may assist in the prevention of other altitude-related diseases, ultimately aiding travelers in adapting to high-altitude environments. However, further research is needed to verify the molecular mechanisms we have identified. Although this study found that *F. duncaniae* and 2-ketoglutaric acid have beneficial effects in reducing intestinal barrier damage caused by high-altitude exposure in mouse models, the complexity and diversity of the human gut microbiota, coupled with the limitations of animal models in fully simulating human intestinal diseases induced by high-altitude exposure, underscore the need for extensive human trials. Therefore, numerous clinical studies are needed to determine whether these agents exert similar beneficial effects on intestinal diseases resulting from high-altitude exposure in humans.

## Figures and Tables

**Figure 1 nutrients-17-01380-f001:**
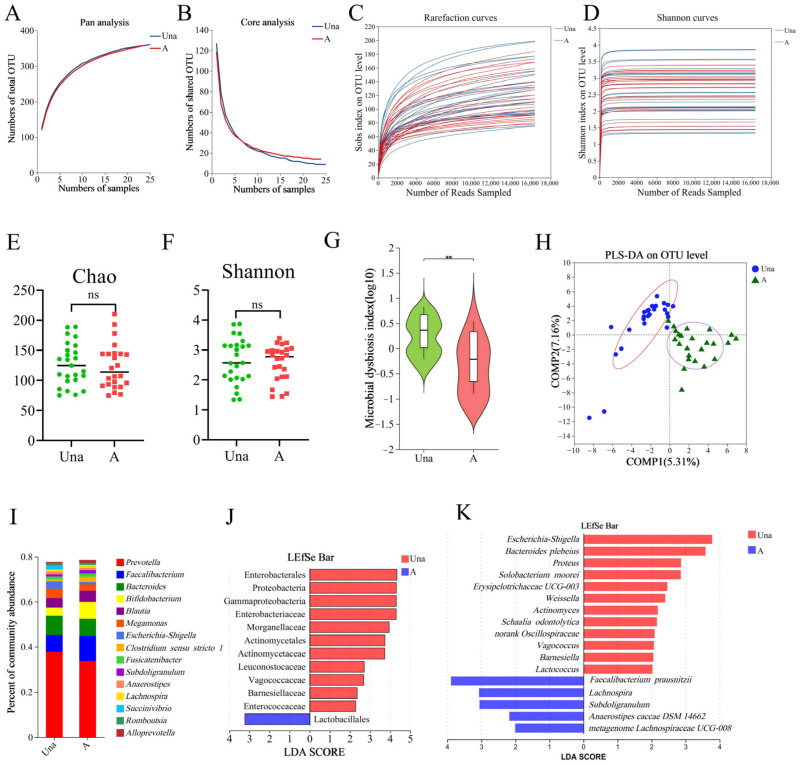
Differences in gut microbiota between volunteers during the acute response and acclimatization phases at high altitude. (**A**,**B**) Pan/Core analysis. The Pan species curve (**A**) and Core species curve (**B**) illustrate changes in total and core species as the sample size increases. The flatness of the pan/core species curve helps determine the adequacy of the sequencing sample size. (**C**,**D**) Rarefaction curves based on the Sobs index (**C**) and Shannon index (**D**) provide valuable insights into the sufficiency of the sequencing data. A leveling off of the curve suggests that the sequencing data are adequate. (**E**,**F**) Comparison of alpha diversity between the Una and A groups, based on Chao’s index (richness) (**E**) and Shannon’s index (diversity) (**F**). (**G**) Comparison of the microbial dysbiosis index between the Una and A groups. (**H**) Partial Least Squares Discriminant Analysis. Each point represents an individual sample, with points sharing the same color and shape belonging to the same group. (**I**) Species composition of the Una and A groups at the genus level, with the bar chart displaying the top 15 genera by relative abundance. (**J**,**K**) Results of LEfSe analysis highlight significantly different bacterial taxa between the groups. The red color indicates a significant increase in the Una group, while blue indicates a significant increase in the A group. Statistical analyses were conducted using Student’s *t*-test (**E**,**F**) or the Wilcoxon rank-sum test (**G**), with ** indicating *p* < 0.01 and “ns” indicating no significant difference.

**Figure 2 nutrients-17-01380-f002:**
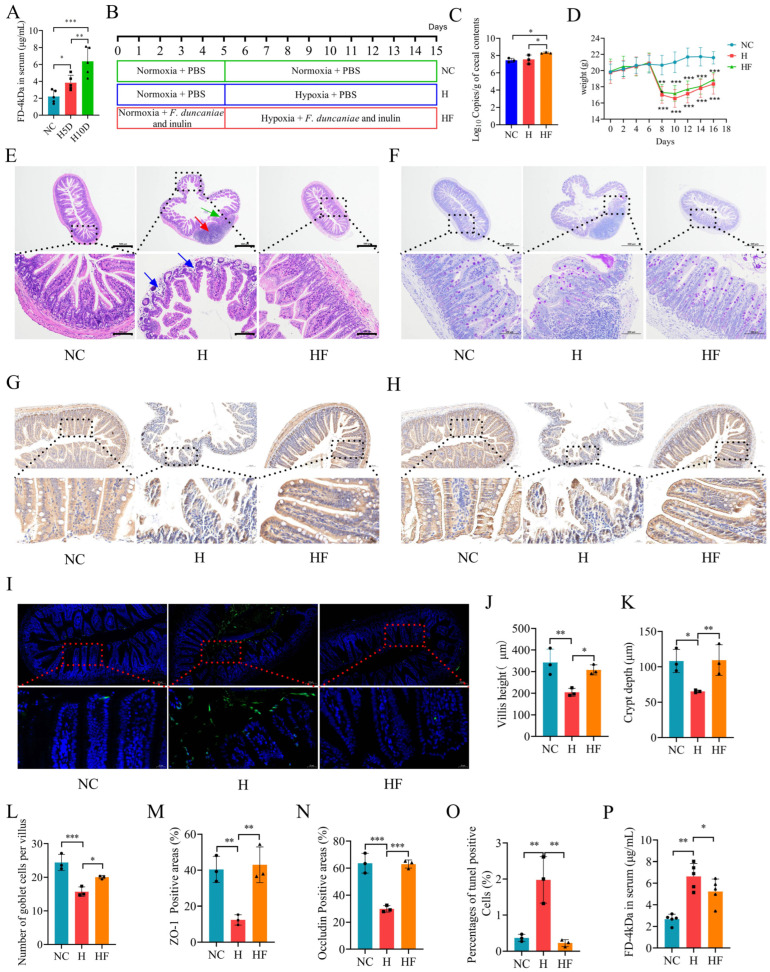
*F. duncaniae* mitigates intestinal barrier damage induced by high-altitude exposure. (**A**) Serum levels of the intestinal permeability tracer FITC-dextran were measured using a fluorescence spectrophotometer (*n* = 5). (**B**) Flowchart illustrating the animal treatment protocol. (**C**) Quantification of *F. duncaniae* in cecal contents via real-time PCR (*n* = 3). (**D**) Body weight changes over the 15-day experimental period (*n* = 8). Statistically significant differences were observed between the NC and H groups, as well as between the NC and HF groups. (**E**) Representative histological sections of ileum tissue stained with H&E at 40× and 200× magnification (*n* = 3). (**F**) Representative histological sections of ileum tissue stained with PAS at 40× and 200× magnifications (*n* = 3). (**G**,**H**) Immunohistochemical analysis of ZO-1 expression at both 10× and 40× magnifications (*n* = 3) (**G**) and occludin expression at the same magnifications (*n* = 3) (**H**). (**I**) Representative images of TUNEL staining in ileal tissues at both 10× and 40× magnifications (*n* = 3). (**J**) Measurement of villus height in the ileum using IMAGEJ (*n* = 3). (**K**) Measurement of crypt depth in the ileum using IMAGEJ (*n* = 3). (**L**) Quantification of goblet cell numbers in the ileum using IMAGEJ (*n* = 3). (**M**,**N**) Percentage of positive area for ZO-1 (**M**) and occludin (**N**) (*n* = 3). (**O**) Percentage of positive cells in TUNEL-stained ileal tissues (*n* = 3). (**P**) Serum levels of the intestinal permeability tracer FITC-dextran were measured using a fluorescence spectrophotometer (*n* = 5). Statistical analysis was conducted using one-way ANOVA (**A**,**C**,**D**,**J**–**P**). * *p* < 0.05, ** *p* < 0.01, *** *p* < 0.001. Data are presented as means ± SD.

**Figure 3 nutrients-17-01380-f003:**
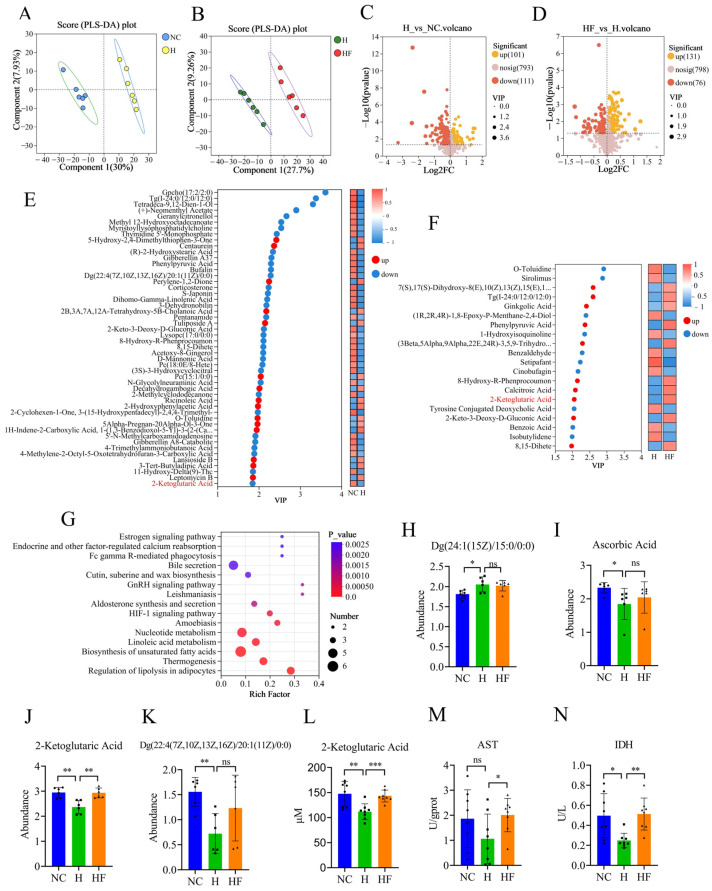
Influence of *F. duncaniae* on the metabolome of cecal contents. (**A**,**B**) PLS-DA analysis was performed to compare the NC and H groups (**A**), as well as between the H and HF groups (*n* = 6) (**B**). (**C**,**D**) Volcano plots illustrate the differential metabolites between the H and NC groups (**C**) and between the HF and H groups (*n* = 6) (**D**). (**E**,**F**) VIP score analysis, based on the weighted coefficients of the OPLS-DA model, ranks the contributions of metabolites to the differentiation between the NC and H groups (**E**), as well as between the H and HF groups (*n* = 6) (**F**). (**G**) KEGG pathway enrichment analysis was conducted using the differential metabolites identified between the NC and H groups (*n* = 6). (**H**–**K**) The abundances of differential metabolites associated with the HIF-1 signaling pathway were evaluated between the NC and H groups (*n* = 6). (**L**) The quantification of 2-ketoglutaric acid was performed at a total protein concentration of 1 g/L of the cecal contents across different groups (*n* = 8). (**M**) The activity of AST in the cecal contents was measured across various groups (*n* = 8). (**N**) The activity of IDH at a total protein concentration of 1 g/L of the cecal contents was analyzed across different groups (*n* = 8). Statistical analyses were performed using Student’s *t*-test (**H**–**N**). Significance levels were defined as follows: * *p* < 0.05, ** *p* < 0.01, *** *p* < 0.001, and “ns” indicates no significance.

**Figure 4 nutrients-17-01380-f004:**
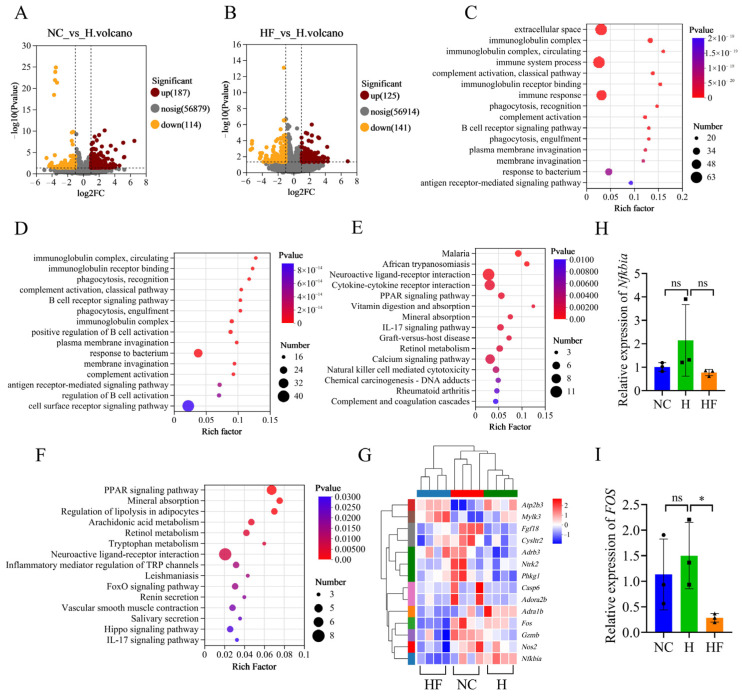
Influence of *F. duncaniae* on the transcriptome of ileal tissues. (**A**,**B**) Volcano plots depict the differentially expressed genes (DEGs) between the NC and H groups (**A**), as well as between the HF and H groups (*n* = 4) (**B**). (**C**,**D**) GO pathway enrichment analyses were performed using the DEGs from the comparisons of the NC and H groups (**C**) and the HF and H groups (*n* = 4) (**D**). (**E**,**F**) KEGG pathway enrichment analyses were conducted for the DEGs in both the NC versus H groups (**E**) and HF versus H groups (*n* = 4) (**F**). (**G**) Heat maps illustrate the DEGs associated with the Apoptosis and Calcium Signaling Pathways (*n* = 4). (**H**,**I**) RT-qPCR validation of the expression levels of *Nfkbia* and *Fos* (*n* = 4) was carried out. Statistical analyses were performed using one-way ANOVA (**H**,**I**), with significance levels defined as follows: * *p* < 0.05, and “ns” indicates no significance.

**Figure 5 nutrients-17-01380-f005:**
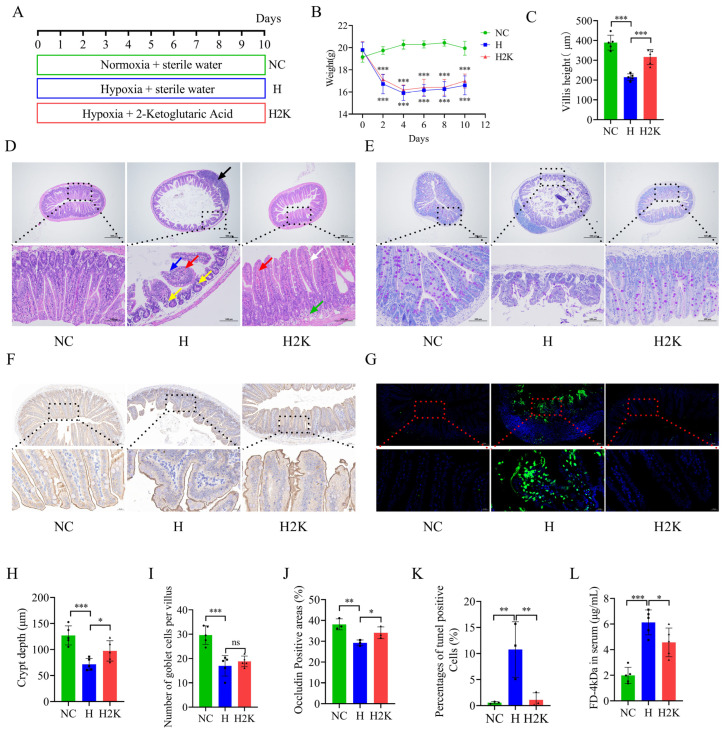
2-Ketoglutaric acid helps to repair intestinal barrier damage in mice resulting from high-altitude exposure. (**A**) Flowchart illustrating the animal treatment protocol. (**B**) Body weight changes over the 15-day experimental period (*n* = 8). Statistically significant differences were observed between the NC and H groups, as well as between the NC and H2K groups. (**C**) Measurement of villus height in the ileum using IMAGEJ (*n* = 5). (**D**) Representative histological sections of ileum tissue stained with H&E at 40× and 200× magnifications (*n* = 5). (**E**) Representative histological sections of ileum tissue stained with PAS at 40× and 200× magnifications (*n* = 5). (**F**) Immunohistochemical analysis of occludin expression at both 10× and 40× magnifications (*n* = 3). (**G**) Representative images of TUNEL staining in ileal tissues at both 10× and 40× magnifications (*n* = 3). (**H**) Measurement of crypt depth in the ileum using IMAGEJ (*n* = 5). (**I**) Quantification of goblet cell numbers in the ileum using IMAGEJ (*n* = 5). (**J**) Percentage of positive area for occludin (*n* = 3). (**K**) Percentage of positive cells in TUNEL-stained ileal tissues (*n* = 3). (**L**) Serum levels of the intestinal permeability tracer FITC-dextran were measured using a fluorescence spectrophotometer (*n* = 5). Statistical analysis was conducted using one-way ANOVA (**B**,**C**,**H**–**L**). * *p* < 0.05, ** *p* < 0.01, *** *p* < 0.001, and “ns” indicates no significance. Data are presented as means ± SD.

## Data Availability

16S rRNA gene sequence data and RNA-seq data were submitted to the NCBI Sequence Read Archive database with the accession numbers PRJNA1206295 (accessible at https://www.ncbi.nlm.nih.gov/bioproject/PRJNA1206295/, accessed on 16 April 2025) and PRJNA1229767 (accessible at https://www.ncbi.nlm.nih.gov/bioproject/PRJNA1229767/, accessed on 16 April 2025), respectively. Untargeted lipidomics data were deposited in the EMBL-EBI MetaboLights database with the identifier MTBLS12075.

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
