# Peer review of "Faecalibacterium duncaniae Mitigates Intestinal Barrier Damage in Mice Induced by High-Altitude Exposure by Increasing Levels of 2-Ketoglutaric Acid"

_nutrients, 2025, doi:10.3390/nu17081380_

Round 1

Reviewer 1 Report

Comments and Suggestions for Authors

The present study highlights probiotic strains that improve intestinal health during high-altitude acclimatization, focusing on Faecalibacterium duncaniae and its protective effects against gastrointestinal damage. The research is relevant and addresses the research question. However, I have a few comments before accepting it for publication.

  1. The author should follow the binomial nomenclature system for bacterial names.
  2. The author should add a reference for lines 622-623.
  3. The author should discuss the results in more detail.
  4. The author expresses the findings of an immunohistochemical analysis of ZO-1 and occludin expression in the positive area (%). Figures 2G and H show a more significant gap in the image (without cells), which may impact the positive area. The author should also indicate the magnification used for this analysis in the figure.
Comments on the Quality of English Language

The English can be improved to express the research more clearly.

Author Response

April 6, 2025

Senior Reviewer

Nutrients

Dear Reviewer, 

Thank you for your consideration and for the valuable comments provided regarding our manuscript entitled “Faecalibacterium duncaniae mitigates intestinal barrier damage in mice induced by high-altitude exposure by increasing levels of 2-ketoglutaric acid” (nutrients-3559680). We greatly appreciate these insightful remarks, which have significantly aided us in revising and enhancing our paper, as well as providing important guidance for our research. We have carefully reviewed the comments and made corrections that we hope will meet your approval. The revised portions are highlighted in yellow in the revised Manuscript. We would like to bring to your attention that the line numbers may change when the Word version of our submitted manuscript is converted to PDF format by the submission system. This could potentially cause inconvenience while you review our revised manuscript. If possible, we recommend reading the Word version for greater ease. Below are the main corrections made in the paper, along with our responses to the reviewers’ comments:

Comments 1: The author should follow the binomial nomenclature system for bacterial names.

Response 1:

We would like to express our gratitude to the reviewers for their valuable and professional suggestions. In the revised manuscript, all bacterial names have been modified in accordance with the binomial nomenclature system (lines 98, 114, 509; Figure 1I-K). Additionally, we have excluded significantly differential species labeled as “Unclassified” from the names presented (Figure 1K).

Comments 2:The author should add a reference for lines 622-623.

Response 2:

Thank you very much for the reviewers’ kind reminders. We have added the relevant references to this content in the revised manuscript (line 530).

Relevant references

31.X. Wang, Y. Li, X. Wang, R. Wang, Y. Hao, F. Ren, P. Wang and B Fang, Faecalibacterium prausnitzii Supplementation Prevents Intestinal Barrier Injury and Gut Microflora Dysbiosis Induced by Sleep Deprivation, Nutrients, 2024, 16, 1100.

32.R. Wang, X. Yang, J. Liu, F. Zhong, C. Zhang, Y. Chen, T. Sun, C. Ji and D. Ma, Gut microbiota regulates acute myeloid leukaemia via alteration of intestinal barrier function mediated by butyrate, Nat Commun, 2022, 13, 2522.

Comments 3: The author should discuss the results in more detail.

Response 3:

The authors appreciate the reviewer’s suggestions and have elaborated on the discussion of the results. Additional content has been included in the discussion and conclusion sections as follows:

  1. Additionally, we observed a significant increase in the relative abundance of several other bacterial species, such as Lachnospira, Subdoligranulum, and Anaerostipes caccaeDSM 14662, during the acclimatization phase compared to the acute response phase. Previous studies have indicated that Subdoligranulum is beneficial for alleviating necrotizing enterocolitis [36], while Anaerostipes caccae is recognized as an important butyrate-producing bacterium [37]. This information suggests that studying the synergistic effects of these bacteria and duncaniae in alleviating intestinal diseases induced by high altitude exposure holds great potential for future applications (lines 521-528).

  1. Due to its multiple functions, 2-ketoglutaric acid is recognized as a safe nutritional supplement in clinical applications, exhibiting protective effects against various diseases, including aging, muscle mass loss, osteoporosis, neurodegenerative diseases, and cardiovascular diseases [41]. However, the role of 2-ketoglutaric acid in treating gastrointestinal disorders induced by high-altitude exposure has not been previously established. The results of this study suggest that 2-ketoglutaric acid holds significant potential as a nutritional supplement for the treatment or prevention of gastrointestinal disorders associated with high-altitude exposure. (lines 543-549).

  1. Interestingly, 2-ketoglutaric acid, known as an immune nutrient factor, plays a crucial role in general immune metabolism. As a glutamine homologue, 2-ketoglutaric acid possesses immuno-enhancing properties, maintaining the gut barrier, increasing the activity of immune cells and neutrophils, promoting phagocytosis, and reducing bacterial translocation in vivo [42]. We hypothesize that duncaniaemay modulate mouse immunity by restoring ketoglutaric acid levels, which could be beneficial in protecting the intestinal barrier from damage caused by high-altitude exposure (lines 559-565).

  1. Although this study found that duncaniaeand 2-ketoglutaric acid have beneficial effects in reducing intestinal barrier damage caused by high altitude exposure in mouse models, the complexity and diversity of the human gut microbiota, coupled with the limitations of animal models in fully simulating human intestinal diseases induced by high-altitude exposure, underscore the necessity for extensive human trials. Therefore, numerous clinical studies are needed to determine whether these agents exert similar beneficial effects on intestinal diseases in humans resulting from high altitude exposure (lines 596-602).

Comments 4: The author expresses the findings of an immunohistochemical analysis of ZO-1 and occludin expression in the positive area (%). Figures 2G and H show a more significant gap in the image (without cells), which may impact the positive area. The author should also indicate the magnification used for this analysis in the figure.

Response 4:

We appreciate the reviewers’ comments. Although Figures 2G and H display a more significant gap, these gaps were excluded from the calculation of the positive area. The positive areas were quantified using Aipathwell software, which automatically identifies cells and tissues while excluding gaps. The magnification for the images is indicated in the figure legend (lines 823-825, 880-882).

We have made every effort to enhance the manuscript and have implemented some revisions. These changes do not affect the key results or the overall framework of the paper. We have highlighted these modifications in yellow in the revised manuscript. We sincerely appreciate the warm and professional efforts of the Reviewer, and we hope that these corrections will receive your approval. If you have any further concerns, please do not hesitate to contact us. Once again, we thank you very much for your comments and suggestions.

Sincerely,

Jingming Jia, Ph.D.

Professor, School of Traditional Chinese Medicine

Shenyang Pharmaceutical University

Shenyang, China 110016

Email: jiajingming@163.com

ORCID: 0009-0002-0519-0854

Reviewer 2 Report

Comments and Suggestions for Authors

The study delves into new territory by examining the role of probiotics in addressing gastrointestinal issues caused by high altitude exposure. With the growing focus on gut microbiota and probiotics, this research provides valuable insight into how specific strains may help alleviate damage caused by hypoxia.Methodology includes both human microbiome analysis amd  mouse developed model to simulate intestinal damage, immunohistochemistry, and transcriptome analysis. The identification of Faecalibacterium prausnitzii and F. duncaniae is significant.Yet, the restoration of ZO-1 and occludin expression, combined with reduction of apoptosis in ileal tissues, presents promising therapeutic opportunities for probiotic-based therapies.The translation from mouse models to humans requires more extended discussion, especially given the complexity and diversity of the human gut microbiota.The potential of 2-ketoglutaric acid to reduce damage is promising, but the paper lacks sufficient detail on its biological significance or how F. duncaniae affects its levels. A more in-depth exploration of the metabolic pathways involved would help clarify its significance.Please also write in italics all bacterial genus and species following international nomenclature .

Author Response

April 6, 2025

Senior Reviewer

Nutrients

Dear Reviewer, 

Thank you for your consideration and for the valuable comments provided regarding our manuscript entitled “Faecalibacterium duncaniae mitigates intestinal barrier damage in mice induced by high-altitude exposure by increasing levels of 2-ketoglutaric acid” (nutrients-3559680). We greatly appreciate these insightful remarks, which have significantly aided us in revising and enhancing our paper, as well as providing important guidance for our research. We have carefully reviewed the comments and made corrections that we hope will meet your approval. The revised portions are highlighted in yellow in the revised Manuscript. We would like to bring to your attention that the line numbers may change when the Word version of our submitted manuscript is converted to PDF format by the submission system. This could potentially cause inconvenience while you review our revised manuscript. If possible, we recommend reading the Word version for greater ease. Below are the main corrections made in the paper, along with our responses to the reviewers’ comments:

Comments 1: The translation from mouse models to humans requires more extended discussion, especially given the complexity and diversity of the human gut microbiota.

Response 1:

The authors appreciate the reviewer’s suggestions regarding this significant issue. We have expanded the discussion on this topic in the revised manuscript. The following additions have been made:

Although this study found that F. duncaniae and 2-ketoglutaric acid have beneficial effects in reducing intestinal barrier damage caused by high altitude exposure in mouse models, the complexity and diversity of the human gut microbiota, coupled with the limitations of animal models in fully simulating human intestinal diseases induced by high-altitude exposure, underscore the necessity for extensive human trials. Therefore, numerous clinical studies are needed to determine whether these agents exert similar beneficial effects on intestinal diseases in humans resulting from high altitude exposure (lines 596-602).

Comments 2: The potential of 2-ketoglutaric acid to reduce damage is promising, but the paper lacks sufficient detail on its biological significance or how F. duncaniae affects its levels. A more in-depth exploration of the metabolic pathways involved would help clarify its significance.

Response 2:

We would like to express our gratitude to the reviewers for their insightful comments. 2-ketoglutaric acid, a key metabolic intermediate in the tricarboxylic acid cycle, plays a significant role in mitigating intestinal injury. Its mechanisms of action include metabolic regulation, antioxidant activity, anti-inflammatory responses, immune regulation, cellular repair, and the modulation of the intestinal microenvironment. The following is a detailed overview of its biological significance:

  1. Antioxidant and Oxidative Stress Regulation

2-ketoglutaric acid can directly or indirectly enhance the activity of antioxidant systems, such as glutathione, neutralize reactive oxygen species, and mitigate oxidative stress-induced damage to intestinal epithelial cells [1]. It can upregulate the expression of antioxidant enzymes, including superoxide dismutase and catalase, thereby enhancing the antioxidant defense capacity of cells [2].

  1. Anti-inflammatory and Immune Regulatory Effects

2-ketoglutaric acid can reduce the release of pro-inflammatory factors, such as TNF-α, IL-6, and IL-1β, which may help alleviate intestinal inflammation [3]. Additionally, it promotes the transformation of macrophages from the pro-inflammatory M1 phenotype to the anti-inflammatory M2 phenotype, thereby reducing the pathological damage associated with inflammatory bowel disease [4]. Notably, recognized as an immune nutrient, 2-ketoglutaric acid plays a crucial role in overall immune metabolism. As a homologue of glutamine, it exhibits immuno-enhancing properties by maintaining the integrity of the gut barrier, increasing the activity of immune cells and neutrophils, promoting phagocytosis, and reducing bacterial translocation in vivo [5].

  1. Promotion of Repair and Regeneration of Intestinal Epithelial Cells

As an intermediate in the tricarboxylic acid cycle, 2-ketoglutaric acid provides energy for the proliferation and migration of intestinal epithelial cells through ATP generation [6]. Furthermore, it restores barrier function by regulating endoplasmic reticulum stress and activating Wnt/β-catenin-mediated proliferation and differentiation of intestinal stem cells [7].

4.Maintenance of Intestinal Barrier Integrity

2-ketoglutaric acid can upregulate the expression of tight junction proteins, including claudin-3, claudin-7, ZO-1, and MLCK. This process reduces intestinal permeability and prevents the translocation of bacteria and toxins [8].

  1. Regulation of Intestinal Flora Balance

2-ketoglutaric acid may serve as a carbon source for certain probiotics, such as Lactobacillus, thereby promoting their growth [9] and inhibiting the colonization of pathogenic bacteria, including Citrobacter [8].

The biological significance of 2-ketoglutaric acid discussed above is outlined in the discussion section (lines 534-565). However, if this study could further investigate and elucidate the potential connections between 2-ketoglutaric acid and various metabolic pathways identified herein—such as the PPAR signaling pathway, mineral absorption, regulation of lipolysis in adipocytes, and calcium signaling pathways, in addition to the differential gene Fos—it would indeed represent a more comprehensive investigation. Unfortunately, due to technical limitations, we plan to explore these areas in future research.

Regarding how F. duncaniae affects ketoglutaric acid levels, this study contains relevant content to address this issue. In the results section, we state: “Furthermore, we measured the activity levels of AST and IDH in the cecal contents. The results indicated that AST activity in the HF group was significantly higher than that in the H group, and IDH activity was significantly elevated in both the HF and NC groups compared to the H group (Figure 3M, N).” (lines 428-431) In the discussion section, we elaborate: “Additionally, to explore the mechanism by which F. duncaniae restores 2-ketoglutaric acid levels, we examined KEGG databases and found that it can express AST and IDH enzymes. Both of these enzymes catalyze specific substrates to produce 2-ketoglutaric acid. Our analysis demonstrated that supplementation with F. duncaniae significantly enhanced the activity of these two enzymes in cecal contents.” (lines 550-554)

  • Chen, Q., Gao, L., Li, J., Yuan, Y., Wang, R., Tian, Y., & Lei, A. (2022). α-Ketoglutarate Improves Meiotic Maturation of Porcine Oocytes and Promotes the Development of PA Embryos, Potentially by Reducing Oxidative Stress through the Nrf2 Pathway. Oxidative medicine and cellular longevity, 2022, 7113793. https://doi.org/10.1155/2022/7113793
  • Murugesan, V., & Subramanian, P. (2003). Enhancement of circulatory antioxidants by alpha-ketoglutarate during sodium valproate treatment in Wistar rats. Polish journal of pharmacology, 55(1), 31–36.
  • Xu, H. W., Fang, X. Y., Liu, X. W., Zhang, S. B., Yi, Y. Y., Chang, S. J., Chen, H., & Wang, S. J. (2023). α-Ketoglutaric acid ameliorates intervertebral disk degeneration by blocking the IL-6/JAK2/STAT3 pathway. American journal of physiology. Cell physiology, 325(4), C1119– https://doi.org/10.1152/ajpcell.00280.2023
  • Li, M., Chen, Q., Zhou, M., Li, X., Wang, Z., & Wang, J. (2025). α-Ketoglutaric Acid Reprograms Macrophages by Altering Energy Metabolism to Promote the Regeneration of Small-Diameter Vascular Grafts. ACS biomaterials science & engineering, 11(1), 518–530. https://doi.org/10.1021/acsbiomaterials.4c01702
  • Wu, N., Yang, M., Gaur, U., Xu, H., Yao, Y., & Li, D. (2016). Alpha-Ketoglutarate: Physiological Functions and Applications. Biomolecules & therapeutics, 24(1), 1–8. https://doi.org/10.4062/biomolther.2015.078
  • Meng, X., Liu, H., Peng, L., He, W., & Li, S. (2022). Potential clinical applications of alpha‑ketoglutaric acid in diseases (Review). Molecular medicine reports, 25(5), 151. https://doi.org/10.3892/mmr.2022.12667
  • Si, X., Song, Z., Liu, N., Jia, H., Liu, H., & Wu, Z. (2022). α-Ketoglutarate Restores Intestinal Barrier Function through Promoting Intestinal Stem Cells-Mediated Epithelial Regeneration in Colitis. Journal of agricultural and food chemistry, 70(43), 13882–13892. https://doi.org/10.1021/acs.jafc.2c04641
  • Wu, D., Fan, Z., Li, J., Zhang, Y., Xu, Q., Wang, L., & Wang, L. (2022). Low Protein Diets Supplemented With Alpha-Ketoglutarate Enhance the Growth Performance, Immune Response, and Intestinal Health in Common Carp (Cyprinus carpio). Frontiers in immunology, 13, 915657. https://doi.org/10.3389/fimmu.2022.915657
  • Si, X., Jia, H., Liu, N., Li, J., Pan, L., Wang, J., & Wu, Z. (2022). Alpha-Ketoglutarate Attenuates Colitis in Mice by Increasing Lactobacillus Abundance and Regulating Stem Cell Proliferation via Wnt-Hippo Signaling. Molecular nutrition & food research, 66(10), e2100955. https://doi.org/10.1002/mnfr.202100955

Comments 3: Please also write in italics all bacterial genus and species following international nomenclature.

Response 3:

We would like to express our gratitude to the reviewers for their valuable and professional suggestions. In the revised manuscript, all bacterial names have been modified in accordance with the international nomenclature (lines 98, 114, 509; Figure 1I-K). Furthermore, we have excluded significantly different species labeled as “Unclassified” from the presented names (Figure 1K). Additionally, all bacterial genera and species have been written in italics (lines 111-112, 128, 347, 698, 705-706, 713, 720, 732-735, 742-745).

We have made every effort to enhance the manuscript and have implemented some revisions. These changes do not affect the key results or the overall framework of the paper. We have highlighted these modifications in yellow in the revised manuscript. We sincerely appreciate the warm and professional efforts of the Reviewer, and we hope that these corrections will receive your approval. If you have any further concerns, please do not hesitate to contact us. Once again, we thank you very much for your comments and suggestions.

Sincerely,

Jingming Jia, Ph.D.

Professor, School of Traditional Chinese Medicine

Shenyang Pharmaceutical University

Shenyang, China 110016

Email: jiajingming@163.com

ORCID: 0009-0002-0519-0854

Reviewer 3 Report

Comments and Suggestions for Authors

The study is interesting and depicts the potential effect of Probiotic Faecalibacterium duncaniae and prebiotic inulin to prevent the intestinal barrier damage provoked by increasing levels of 2-Ketoglutaric Acid. However, I have some concerns that need to be addressed:

  1. Write the full name of abbreviations as they appeared first in the text such as Fos (Line 26), CD1d, IL-17 (Line 67).
  2. All the key words are merely replication of the Title, replace all the key words with other suitable and relevant keywords.
  3. Replace the word “transplanted” from the sentence “The identified probiotic was subsequently transplanted into the……. high altitude exposure (Lines 98-100).
  4. Where is Figure 2B (Line 173), similarly suddenly figure 5 (Line 184) is popping. Arrange all the figure numbers appropriately in the entire manuscript.
  5. FITC-dextran was calculated using the standard curve (Lines 247-248), what standard curve was used, put the line of regression equation or the logarithmic equation used to quantify FITC-dextran.
  6. In Figure 2 D it’s not clear that the statistical difference was calculated between which groups. Mention a clear statement in the figure legend.
  7. For the histological images depicted in Figure 2E, the scale bar is not visible. Construct it again for clear visibility.

Author Response

April 6, 2025

Senior Reviewer

Nutrients

Dear Reviewer, 

Thank you for your consideration and for the valuable comments provided regarding our manuscript entitled “Faecalibacterium duncaniae mitigates intestinal barrier damage in mice induced by high-altitude exposure by increasing levels of 2-ketoglutaric acid” (nutrients-3559680). We greatly appreciate these insightful remarks, which have significantly aided us in revising and enhancing our paper, as well as providing important guidance for our research. We have carefully reviewed the comments and made corrections that we hope will meet your approval. The revised portions are highlighted in yellow in the revised Manuscript. We would like to bring to your attention that the line numbers may change when the Word version of our submitted manuscript is converted to PDF format by the submission system. This could potentially cause inconvenience while you review our revised manuscript. If possible, we recommend reading the Word version for greater ease. Below are the main corrections made in the paper, along with our responses to the reviewers’ comments:

Comments 1: Write the full name of abbreviations as they appeared first in the text such as Fos (Line 26), CD1d, IL-17 (Line 67).

Response 1:

We sincerely appreciate the reviewer’s suggestions. In the revised manuscript, we have provided the full names of the relevant terms: FBJ osteosarcoma oncogene (Fos) (lines 53-54), Cluster of Differentiation 1D (CD1d)(line 96), and Interleukin-17 (IL-17) (line 97).

Comments 2: All the key words are merely replication of the Title, replace all the key words with other suitable and relevant keywords.

Response 2:

We sincerely appreciate the reviewer’s suggestions. In the revised manuscript, we have updated several keywords, which now include: gut microbiota, 2-ketoglutaric acid, probiotics, gastrointestinal issues, and hypoxia exposure (lines 59-60).

Comments 3: Replace the word “transplanted” from the sentence “The identified probiotic was subsequently transplanted into the……. high altitude exposure (Lines 98-100).

Response 3:

We sincerely appreciate the reviewer’s suggestions. In the revised manuscript, we have replaced the term “transplanted.” The original sentence, “The identified probiotic was subsequently transplanted into the intestines of mice to evaluate its protective effects against intestinal barrier damage induced by high altitude exposure,” has been revised to: “ The identified probiotic was subsequently administered via gavage to the intestines of mice to evaluate its protective effects against intestinal barrier damage induced by high altitude exposure.”(line 124)

Comments 4: Where is Figure 2B (Line 173), similarly suddenly figure 5 (Line 184) is popping. Arrange all the figure numbers appropriately in the entire manuscript.

Response 4:

We would like to express our gratitude to the reviewers for their valuable comments. The issue may have arisen because we referenced Figure 2B and Figure 5A in the methods section, which caused the journal submission system to incorrectly position the image that should have appeared in the results section during the conversion from the Word document to PDF format. This error occurred when the journal submission system arranged the images. We have removed the reference to Figure 2B (line 191) and Figure 5A (line 202) in the methods section of the revised manuscript, ensuring that it is cited solely in the results section. We believe this issue has been resolved.

Comments 5: FITC-dextran was calculated using the standard curve (Lines 247-248), what standard curve was used, put the line of regression equation or the logarithmic equation used to quantify FITC-dextran.

Response 5:

We would like to express our sincere gratitude to the reviewers for their valuable comments. In the revised manuscript, we have included the regression equation line (line 245).

Comments 6: In Figure 2 D it’s not clear that the statistical difference was calculated between which groups. Mention a clear statement in the figure legend.

Response 6:

Thank you very much for your reminder. We sincerely apologize for this oversight. We have indicated in the legend that the statistical differences are observed between the NC and H groups, as well as between the NC and HF groups (lines 818-819).

Comments 7: For the histological images depicted in Figure 2E, the scale bar is not visible. Construct it again for clear visibility.

Response 7:

We would like to express our gratitude to the reviewers for their valuable comments. To address this issue, we have redrawn the figure, making the scale bar bolder to enhance visibility (Figure 2E).

We have made every effort to enhance the manuscript and have implemented some revisions. These changes do not affect the key results or the overall framework of the paper. We have highlighted these modifications in yellow in the revised manuscript. We sincerely appreciate the warm and professional efforts of the Reviewer, and we hope that these corrections will receive your approval. If you have any further concerns, please do not hesitate to contact us. Once again, we thank you very much for your comments and suggestions.

Sincerely,

Jingming Jia, Ph.D.

Professor, School of Traditional Chinese Medicine

Shenyang Pharmaceutical University

Shenyang, China 110016

Email: jiajingming@163.com

ORCID: 0009-0002-0519-0854

Round 2

Reviewer 2 Report

Comments and Suggestions for Authors

The paper is significantly improved and merit publication